# "We don't routinely check vaccination background in adults": a national qualitative study of barriers and facilitators to vaccine delivery and uptake in adult migrants through UK primary care

Jessica Carter [1], Anushka Mehrotra,[1] Felicity Knights,[1] Anna Deal,[1,2] Alison F Crawshaw [1], Yasmin Farah,[1] Lucy Pollyanna Goldsmith [3], Fatima Wurie,[4] Yusuf Ciftci,[5] Azeem Majeed,[6] Sally Hargreaves [1]

JC, AM and FK are joint first authors.

For numbered affiliations see end of article.

**Correspondence to**
Sally Hargreaves;
s.hargreaves@sgul.ac.uk

## ABSTRACT

**Objectives** Explore primary care professionals' views around barriers/facilitators to catch-up vaccination in adult migrants (foreign-born; over 18 years of age) with incomplete/uncertain vaccination status and for routine vaccines to inform development of interventions to improve vaccine uptake and coverage.

**Design** Qualitative interview study with purposive sampling and thematic analysis.

**Setting** UK primary care.

**Participants** 64 primary care professionals (PCPs): 48 clinical-staff including general practitioners, practice nurses and healthcare assistants; 16 administrative-staff including practice managers and receptionists (mean age 45 years; 84.4% women; a range of ethnicities).

**Results** Participants highlighted direct and indirect barriers to catch-up vaccines in adult migrants who may have missed vaccines as children, missed boosters and not be aligned with the UK's vaccine schedule, from both personal and service-delivery levels, with themes including: lack of training and knowledge of guidance among staff; unclear or incomplete vaccine records; and lack of incentivisation (including financial) and dedicated time and care pathways. Adult migrants were reported as being excluded from many vaccination initiatives, most of which focus exclusively on children. Where delivery models existed, they were diverse and fragmented, but included a combination of opportunistic and proactive programmes. PCPs noted that migrants expressed to them a range of views around vaccines, from positivity to uncertainty, to refusal, with specific nationality groups reported as more hesitant about specific vaccines, including measles, mumps and rubella (MMR).

**Conclusions** WHO's new Immunization Agenda 2030 calls for greater focus to be placed on delivering vaccination across the life course, targeting underimmunised groups for catch-up vaccination at any age, and UK primary care services therefore have a key role. Vaccine uptake in adult migrants could be improved through implementing new financial incentives or inclusion of adult migrant

## STRENGTHS AND LIMITATIONS OF THIS STUDY

⇒ A key strength of this study is the number and variety of primary care staff included from across England in diverse settings.
⇒ Interviewees were a self-selecting group, which may have affected the profile of those responding—a common consideration in qualitative research.
⇒ A large number of practices were involved, however, and this diversity and the scale of the study is likely to have added to the breadth of experience and solutions reflected in our findings, as well as enhancing the validity.
⇒ The structure and experience of primary care across Europe and between the devolved nations of the UK may differ so the recruitment only within England may limit the generalisability of the findings, however we note other European and international studies have come to the similar conclusions.

vaccination targets in Quality Outcomes Framework, strengthening care pathways and training and working directly with local community-groups to improve understanding around the benefits of vaccination at all ages.

## INTRODUCTION

Adult migrants in Europe—particularly those from low-income and middle-income countries—may be at risk of underimmunisation for routine vaccinations due to missed vaccines and doses as children (due to lack of availability, war/disruption, poorly functioning health systems, and personal, social and physical barriers to accessing vaccines), and/or missed boosters, and differing vaccination schedules in their home country (especially for newer vaccines such as human

papillomavirus vaccine (HPV)), and so may not be aligned with the UK's vaccination schedule on arrival.[1–3] Additional vaccines may be recommended if they return to their home country, or for specific occupations (eg, tetanus and hepatitis B). Some migrant populations are known to be at risk of underimmunisation[2 4–6] and were involved in recent outbreaks of vaccine-preventable diseases in Europe, including measles.[1] However, adolescent and adult migrants, beyond school age, are often not routinely incorporated into vaccination programmes on arrival to most European countries, including the UK.[7] The COVID-19 pandemic has highlighted shortfalls in engaging older migrants, and other marginalised groups, in vaccination programmes,[8] yet it has also presented a range of new opportunities and innovations in vaccine service delivery and policymaking to these groups, which merit greater consideration beyond the pandemic.

The WHO's new Immunization Agenda 2030 (IA2030)[9] aims to improve vaccine coverage for vaccine-preventable diseases (VPDs), placing an emphasis on achieving equitable access for vulnerable populations and integrating catch-up vaccination for missed vaccines and doses throughout the life course. WHO recommends that all countries have a catch-up vaccination policy and catch-up vaccination schedule in place, to close immunisation gaps that would otherwise compound as populations increase in age,[10 11] and that it is always 'better to vaccinate late than never'. Although age limits apply for administration of a small number of vaccines, for most VPDs, providing vaccines late will still result in protection against morbidity and mortality, as well as reducing transmission and risk of outbreaks, with personal and community-level benefits. Specific WHO guidance for catch-up vaccination is available[10]; in Europe, the European Centre for Disease Prevention and Control has published guidance on catch-up vaccination in children and adult migrants on arrival,[12 13] calling for healthcare providers to consider revaccinating adult migrants with uncertain vaccination status or no recorded history of vaccination. For UK arrivals, advice is available from the UK Health Security Agency (UKHSA) on the 'vaccination of individuals with uncertain or incomplete immunisation status' (see box 1), which will be relevant to most arriving migrants.[14] However, the extent to which these guidelines and policies are put into practice and prioritised by UK primary care—tasked with delivering the majority of the UK's vaccine programmes—is not known. No studies to date have explored the views and experiences of frontline primary care teams on approaches to catch-up adult vaccination in arriving migrants. We therefore did a national qualitative in-depth interview study with a range of primary care professionals (PCPs) to understand the challenges and needs of migrant populations with regards to catch-up vaccinations programmes, and facilitators and solutions to addressing gaps in service provision.

---

**Box 1  Vaccination of individuals with uncertain or incomplete immunisation status. Reproduced from a study conducted by Public Health England[14]**

From 10th birthday onwards:
⇒ Td/IPV and MenACWY* and MMR.
　　Four-week gap.
　　Td/IPV and MMR.
　　Four-week gap.
　　Td/IPV.
　　First booster of Td/IPV—preferably 5 years following completion of primary course. Second booster of Td/IPV—ideally 10 years (minimum 5 years) following first booster.
⇒ HPV:
　　–All women who have been eligible remain so up to their 25th birthday.
　　–Men born on or after 01 September 2006 are eligible up to their 25th birthday.
⇒ Subsequent vaccination—as per UK schedule (see influenza vaccine, shingles vaccine, PPV) and COVID-19.
*Those aged from 10 years up to 25 years who have never received a MenC-containing vaccine should be offered MenACWY. Those aged 10 years up to 25 years may be eligible or may shortly become eligible for MenACWY usually given around 14 years of age. Those born on/after 01 September 1996 remain eligible for MenACWY until their 25th birthday. (Tetanus and diptheria (Td), inactivated polio vaccine (IPV), meningococcal ACWY vaccine (MenACWY), pneumococcal polysacharide vaccine (PPV)).

---

## METHODS
### Design
Qualitative semi-structured interviews of both clinical and administrative staff were undertaken by telephone, following a topic guide collaboratively developed by the research team with support from a board of migrant representatives. The guide was piloted prior to data collection and iteratively developed throughout the data collection process, with the addition of further prompts and probes to develop richer understanding and addressed key areas around approach to vaccination of adult migrants, factors affecting vaccine hesitancy and uptake and possible interventions to strengthen delivery (box 2). The team comprised two general practitioners (GPs) and four academics, and was supported by a wider project board of a diverse group of migrant ambassadors. The range of professional and personal experience supported integration of multiple perspectives throughout the design, collection and analysis stages. The inclusion of two GPs in the research team brought knowledge of UK primary care to the study but required careful reflection during interviews and data analysis and was balanced by the inclusion of non-GP research team members at the interview stage.

### Patient and public involvement
A board of migrant representatives supported the design of this study and development of the topic guides.

**Box 2   Topic guide**

Background questions:
⇒ Proportion of migrants seen at practice, migrant health training and experience.
⇒ General barriers and facilitators to registration and provision of care for migrant patients.

Questions pertaining to vaccination of adult migrants:
⇒ Are you aware of any guidance regarding vaccination and infectious disease screening in migrants?
⇒ Have there been any outbreaks of vaccine preventable diseases or cases of vaccine preventable diseases in your area involving migrants—we are particularly interested in adults? If yes, what do they think the reasons might be?
⇒ What experience have you had with adult migrant patients and vaccination?

Questions regarding practice approach to vaccination of adult migrants:
⇒ How do you approach catch-up vaccination in the adult migrant patient group, specifically ensuring adult migrants are caught up to align with the UK schedule?
⇒ Who is responsible for vaccination at your practice?
⇒ Is there a mechanism at your practice or in your area to engage adult migrants on catch-up vaccination?
⇒ Is there a local catch-up vaccination pathway?
⇒ Do you target any specific groups?

Questions regarding possible interventions to increase uptake of catch-up vaccination in migrants:
⇒ If there are no mechanisms/pathways in place locally do you think there should be?
⇒ What could such a system look like?
⇒ Are you aware of any other interventions relating to vaccination in migrants? If so, what made them successful/unsuccessful?
⇒ What do you think about a migrant health check, and what vaccinations would be important to cover in this for adult migrants in your view?
⇒ What in your opinion would be the key to a successful intervention/behaviour change in primary care?

## Setting

Latest figures show there are 6822 GP practices in England, the majority are in urban environments as are migrant populations. Participants were recruited from 50 GP practices. Fifty (78%) participants were from practices in urban settings and 14 participants (22%) from suburban or rural settings across England. Practices were based in one of six local Clinical Research Networks (CRNs)—CRN Kent, Surrey and Sussex; CRN South London; CRN North Thames; CRN North West London; CRN West Midlands; and CRN Greater Manchester with the exception of a practice in Newcastle and another in Oxford.

## Participants

Participants were purposively sampled to capture the diversity of experiences in general practice, from administrative and clinical primary care roles and practices which varied both in size, and urban/rural location, factors which could influence the number of migrant patients and the organisation of care. Recruitment occurred via local CRNs, 'word of mouth' invitations from colleagues and a number of primary care newsletters, social media groups and practice manager mailing lists. All participants who expressed an interest in taking part were emailed a participant information sheet and consent form and invited to a telephone interview at a time of mutual convenience, with written informed consent being given in advance. Each participant was given £20 vouchers as compensation for their time.

## Data collection

Telephone interviews, between 30 and 60 minutes, were carried out by JC (GP), FK (GP registrar) and AD and AFC (academic researchers) who made field notes in the majority of cases. Interviews were distributed randomly to research team members. Findings from the initial interviews were discussed across the group and led to the development of additional prompts and lines of questioning in the topic guide, as well as additional lines of questioning for non-clinical participants. All but three of the interviews were digitally recorded and transcribed verbatim by professional transcription service. The remaining three were lost through technical error but were typed up from extensive field notes. Transcripts were anonymised with a coded participant number and checked for accuracy. Data collection continued until there was thematic saturation[15] across all core themes as unanimously agreed across the team.

## Data analysis

Data analysis was inductive, based on the stages of thematic analysis.[16] The transcripts were read repeatedly by AM (familiarisation) and emerging themes and patterns were identified and discussed with FK and JC who had also previously immersed themselves in the data. Initially, a coding of 10 transcripts on Microsoft Excel by AM allowed identification of emergent themes and discussion with FK and JC. NVivo (V.13) was then used to organise codes and iteratively refine and develop the emerging coding framework through a process of constant comparison, with close attention paid to non-confirmatory cases which contradicted existing themes. The final coding and themes were conceptualised through recurrent discussion by AM, FK, JC and SH. Active reflexivity was attempted from the study's onset, and input from across the multidisciplinary team, with support from the migrant advisory board, facilitated robust discussion throughout.

## RESULTS

In total, 64 interviews were conducted. Forty-eight interviews were held with primary care staff: 25 GPs, 15 practice nurses, 7 healthcare assistants (HCAs), 1 clinical pharmacist, 11 practice managers and 5 receptionists. Participants were aged between 25 and 74 with a mean age of 45 years old (SD 11.8) and had been working in primary care between 1 and 35 years (mean 12.27 years SD 9.45). The majority of staff (50 (78.1%)) worked in

| Table 1 | Characteristics of participants |
|---|---|
| **Characteristics** | **Total participants (n=64)** |
| Staff type | General practitioners: 25 (39%)<br>Practice nurses: 15 (23.5%)<br>Healthcare assistants: 7 (11%)<br>Pharmacist: n=1<br>Practice managers: 11 (17%)<br>Receptionists: 5 (8%) |
| Ethnicity | African: 4 (6.3%)<br>Other Asian background: 2 (3.1%)<br>Mixed: 3 (4.7%)<br>Other white: 5 (7.8%)<br>Caribbean: 1 (1.6%)<br>Indian: 11 (17.2%)<br>Pakistani: 3 (4.7%)<br>White British: 32 (50%)<br>White Irish: 2 (4.7%) |
| Age | 45 years (SD 11.8 years) |
| Sex | Female: 54 (84.4%)<br>Male: 10 (15. 6%) |

urban practices. Characteristics of included participants are presented in table 1.

Participants had varied exposure of vaccine delivery in migrant patients, but the data were convergent across this breadth of migrant healthcare experience, geographical area and participant profession. The main themes that emerged from data analysis were; the existence of multiple barriers to the delivery of catch-up vaccination to migrant patients, including vaccine acceptance and PCP training; the fragmented nature of adult migrant catch-up vaccination models despite existence of guidelines; the role of travel vaccination and occupational health have in adult migrant catch-up vaccination and next steps for strengthening delivery of catch up vaccination with existence of positive attitudes to strengthening primary care's role through numerous PCP enacted or suggested solutions to barriers given.

## Existence of multiple barriers reported by PCPs to vaccine uptake in adult migrants
### Patient acceptance of vaccines from PCPs
Participants reported that their migrant patients express a range of views around vaccines from positivity to uncertainty, to refusal. Generalised mistrust and misinformation about vaccinations in migrant groups was commonly reported, which was often perceived by PCPs as resistance to information-sharing about the vaccine in question.

> It's really hard to break through that barrier of… this is the evidence [about this vaccine]… I don't think they're listening… they're thinking… this is someone from my community saying this [other information]. And you're not from my community… I don't know if you have the best interests [in mind]. GP10

### Different nationalities have different views on vaccines
Some PCPs gave their views on vaccine acceptance and uptake linked to specific nationalities, and most often reported beliefs or experiences that migrants originating from Eastern Europe, France and Italy, and Somalia and Bangladesh tend to be hesitant about vaccines. Table 2 provides illustrative quotes. Fixed negative views around vaccines were most often reported from Eastern European migrants, who were also viewed as having poor vaccination records and as wanting to follow a different vaccination schedule (as per protocols in their home country), with some returning to their own countries to be vaccinated. The doctor–patient relationship was highlighted

| Table 2 | Perceptions of staff around acceptance and uptake in specific nationality groups |
|---|---|
| **Participant reporting** | **Quote** |
| GP | '…now it's more Bangladeshi, so Somalian was really with the MMR thing. But we still find more Bangladeshi families delaying or refusing the immunisation of their babies……So, yes they always blame… This is too much, the baby is young, we're not sure about the long term effects.' |
| GP | '[The Somalian population]…is a massive concern for us, with regards the patients unfortunately, falsely attributing MMR with an autism link……. I think it was the belief of autism, but why more in the Somali community than any other minority group, I'm not too sure.' |
| HCA | '[The Somalian population are] … very happy to vaccinate as elderly patients. But, [they think]…the children will get something, get over it. And I think with MMR, they do feel that there's side effects. They think that it causes Autism and things like that.' |
| Practice nurse | 'I don't know where, Somalia or Eritrea that there was only one interpreter in London who could speak their language. Even their care worker obviously could not speak their language. And so, trying to get immunisation history or any history out of these two young men was totally impossible.' |
| HCA | 'I would say that Europeans [migrants], they refuse because they think they've had them, even if it's been a long time and they don't know.' |
| GP | 'What I have noticed is that when a patient comes from… Eastern European countries… they do come in with a vaccination record. It's usually incomplete… and sometimes we doubt [it is true and], whether…you can pay someone to give you a vaccination record but it actually hasn't happened.' |

GP, general practitioner; HCA, healthcare assistant .

as a key factor in tackling mistrust and vaccine hesitancy; some PCPs felt this represented a barrier and that it was easier for migrant patients to connect with PCPs from their own communities.

### Language barriers

Language barriers leading to an inability to communicate vaccination histories and understand vaccine offers were felt by participants to reduce the likelihood of migrants accessing catch-up vaccinations, compounded by a lack of written communication in a variety of languages about vaccine services.

> Language can be a barrier for subtleties of communication, despite language line" GP21 "I think we probably ought to translate that communication [about vaccine programmes] in written Bengali, and perhaps Somali as well. GP10

> There's usually a long wait and possibly a language barrier as well that may stop [people] from communicating or trying to make that appointment. PN 15

### Lack of accurate vaccine histories and fear of immigration

Participants raised the fact that unclear or poorly documented vaccination histories meant staff were unclear as to what to do, as well as highlighting problems with vaccination records not being transferred within the National Health Service (NHS), and a lack of availability of records from migrants' home countries, including limited translation of previous records into English. Some migrants were reported as having different ages recorded, leading to challenges determining vaccine eligibility. Issues were raised about immigration status and PCPs reported migrant patient fears about being reported to authorities if migrant patients attended the GP and disclosed country of origin as part of their vaccine history.

> And we're certainly not being given any records from other countries that might support [vaccination catch-up]… unless the patient is super well-organised and providing that it happens to be in English or a language that's directly transferable… Admin 6

> I think immigration status, out of anything, is going to be the main issue. A lot of people that live in this country without status, going to the GP is a massive risk. PN 13

### Lack of training and unawareness of guidelines among PCPs

Health-system and staff barriers to providing catch-up vaccination for adult migrants included lack of training among staff and lack knowledge of guidance around catch-up vaccination.

> The nurses would need some kind of education in how to complete incomplete vaccination programmes in adults. Admin 12

> So, no, I'm not aware of any guidance for [vaccination in] migrant people. GP 2

### Time and financial pressures

There were also a number of additional barriers to accessing care at staff and system level which were felt to reduce the likelihood of adult migrant patients being offered and accepting a catch-up vaccine or travel vaccines through the travel clinic. These included a lack of time to carry out proactive catch-up programmes, or to follow-up on opportunistic or challenging conversations where a vaccination need was highlighted, especially when using a translator. The financial pressures and impact of vaccination programmes falling outside of current incentive schemes, such as Quality Outcomes Framework (QOF), also impacted on the time available for the programmes.

> It's just time pressure, the way that the general practice is working at the moment unfortunately is reactive…And so, with things like vaccinations, especially if it's catch up or screening, can always wait… [because] you're going to deal with [someone's chest infection or…diabetes] before you deal with their symptomatic screening. GP6

> There are "no incentives for catchup vaccination, MMR… especially compared to childhood immunisations and chronic diseases in QOF. (GP16)

The above represent barriers across all vaccinations. There were barriers reported to specific vaccines in the UK schedule and these have been summarised by vaccine in table 3.

### Fragmented models for vaccine delivery to adult migrants

Almost all clinical staff reported the availability of good catch-up programmes for childhood vaccination among recently arrived migrants, with some practice nurses (PNs) specifically quoting the Public Health England Schedule for individuals with uncertain vaccination status. Incentivisation for under-5s vaccination included the QOF, and well-resourced systems to ensure children are not missed, including the vaccination record 'red book' and using recall systems to contact patients, such as sending reminder texts. By contrast, adult migrants were often reported as being excluded from vaccination initiatives. One GP stated that over 5's and adults are sometimes assumed to be 'up to date from the country they come from', and many staff, especially GPs and administrators, were not aware of any catch-up vaccination programmes for adult migrants.

> We don't routinely check vaccination background in adults. GP 16

> We do catch-up vaccinations for children and young adults who've missed their primary vaccinations, but in terms of adults or people who are arriving to the UK, no. Admin 6

> Ad hoc. We haven't had a particular programme for [adult catch up vaccinations]. GP15

Where adult catch-up vaccination was provided, models of delivery were diverse and fragmented, comprising a

**Table 3** Key barriers to adults acquiring specific vaccines

| Disease | Key barrier perceived by healthcare professionals | Quote | Professional |
|---|---|---|---|
| Influenza | Multiple staff involved creates risk of disjointed process | 'Reception staff call…give [clinicians] the list…then the nurses…[and] doctors vaccine them for both child …and adult.' | Admin 13 |
| | Perceived side effects | '[Adults with flu jab] sometimes they don't want it because they said they had it before and they had side effects, so yes, that's the main thing.' | GP 13 |
| | Perceived poor understanding of influenza among migrant patients | 'I find it difficult to convince them that [flu vaccine]… is useful. Because most …[adult migrants] don't understand the concept of flu.' | GP 4 |
| | Low uptake among younger adult migrant patients | 'We find that, generally, the over 65s will take it and under 65s will have very low uptake…[not sure] it makes a difference with what ethnic background they're from…' | GP 16 |
| | Specific health beliefs surrounding influenza vaccine and immune system | '[Adult migrants]… are refusing because they want to have a[immune] system and teach their body to fight against a virus…[or] they had bad side effects.' | HCA 4 |
| Hepatitis A | Requires patient to proactively seek vaccine | 'We do …[this] when a patient contacts us, because they're either worried about hepatitis or they're thinking they're going to travel [to their home country].' | Admin 13 |
| Hepatitis B | Not within NHS catch up vaccination schedule | 'Vaccination is not within the schedule, so it has to be treated like a private prescription…and [can be] occupation[al] [eg nurses from South India].' | GP 4 |
| HPV | Taboo subject for some migrants | 'Doesn't seem to be a very good uptake of it, in the migrant communities that we have…I think an anything remotely to do, within the genitalia area. When you to to discuss that… it's normally a difficult conversation to have with a lot of the migrant families…. I think they find that a bit of a taboo subject… [they] generally come in groups… [with] mum, dad and maybe a couple of children… [which] makes conversations like HPV… more difficult to discuss.' | HCA 6 |
| Meningococcal | Potential for missed opportunities outside of travel-related risk | 'If [adult migrants have] …not had a meningitis, I will always offer that to them as part of the travel thing.' | PN 2 |
| MMR | False link with autism | 'The Somalian population… falsely attributing MMR with an autism link….' | GP 1 |
| | Electronic systems provide excess alerts leading to healthcare professional desensitisation | 'EMIS (an online practice system) is very annoying because every single patient for who it doesn't have MMR date, it says MMR is outstanding…when people have come, especially if they're refugees or asylum seekers, they won't have that paperwork.' | GP 2 |
| | Less perception of need at older age | '[When migrant patients register]…Especially for under 40, we try to find out, the MMRs, if they have them…If they are young they will accept. But then, the standard for patients over 40, they don't want anymore.' | HCA 4 |
| Shingles | Lack of understanding of shingles | 'There's not enough education around it… it's not something as well-known as, say the flu…it's difficult to get [a translator] for every single patient, to educate them what shingles is.so, I think… it's more education that's needed.' | HCA 6 |

GP, general practitioner; HCA, healthcare assistants ; NHS, National Health Service.

range of clinics and providers, different staff members (primarily nurses) and a combination of opportunistic and proactive programmes. Providers of catch-up vaccination for adult migrants included: NHS GP practices, detention centres (for undocumented migrants and asylum seekers), migrant-specific or language-specific clinics, private clinics and specialised clinics (eg, sexual health clinic in China Town), with distinct benefits and challenges.

> Detention centres: Interpreters weren't always readily provided when I was at the detention centres. We found that really difficult and it took several visits [to determine which vaccines were required and these to be given]. PN13

> [on local community infectious disease led clinic] And they have a large Somalian support network there, so they have interpreters, and bits and pieces………. They will go in, and there will be a Somalian phlebotomist and doctor, and so they engage with it that way, much easier. HCA 6

Respondents reported vaccinations programmes were a mix of opportunistic and proactive delivery approaches. Proactive programmes included methods such as setting up searches, call and recall systems to contact patients and targeted campaigns for specific vaccinations (eg, influenza).

> We run recalls [for adult migrant catch-up] constantly throughout the year. We will target separate cohorts of patients, just so we can make sure we're recalling everybody. Admin 5

Opportunistic usually meant identifying a patient needed a vaccination when they were attending the practice for another reasons. The vaccine could be given immediately, or the patient booked into an appointment at a later date.

> …if I notice and if I remember or have time to mention it, then I encourage people to… [but] they're usually coming with quite a few issues, and we're using an interpreter… there's a lot to cover…[hence no time to cover vaccination]. GP 18

There are also diverse approaches to vaccine delivery between practices, with different staff involved in different aspects of the vaccine programme. However, many programmes are nurse-led, with the practice nurse having main responsibility.

[It's a] mixture of me, one of the partners, and then the reception staff are the ones who actually call the patients and arrange for them to come in. Admin 13

If they're struggling to get somebody to agree [to take a specific vaccine]… we get the named GP. to take responsibility for having that conversation and trying to talk them round. Admin 9

…our vaccines are really well-run at the practice by one of the nurses in particular. She runs the whole immunisation program, the childrens, the flu, the catch up, everything. So, I would imagine that there's probably a lot going on that I'm not aware of. I suspect and she always goes on updates and is very much aware of new guidance to things so I'm sure that she's probably doing a lot of stuff behind the scenes that I'm not aware of. GP 3

### Travel vaccination and occupational vaccines

Provision of catch-up vaccines and additional vaccines to adult migrants was also mentioned in the context of travel and occupational requirements. Delivery of travel vaccinations was highlighted by a variety of participants for migrants visiting their home countries and travelling to Haj.

I think people are very good at knowing they need vaccinations, especially people who have been settled in England for quite a long time and are maybe making an infrequent return visit home to may visit relatives or family elders or to go for a celebration. Admin 6

They will go for the bare minimum of what is offered, or what they need to have as certificated. If they're doing the pilgrim to the Haj, then they have to get the meningitis. If they… need yellow fever, they'll get the yellow fever…or they just don't have anything. PN 2

Different nationalities were reported as having varying levels of engagement with travel vaccine uptake. One PN reported Bangladeshi families travelling more being 'more engaged' than Middle Eastern people. Another reported Europeans as 'more engaged with travel clinics than…people… from Pakistan, India, Bangladesh and African countries'. (PN2) African patients were described as having a poorer uptake of travel clinics than Europeans 'people returning to DRC or Tanzania…their uptake is poorer than younger European people' PN2.

Participants noted that travel clinics can also be an opportunity for opportunistic adult catch-up:

The nurses who do the travel clinics are certainly very switched on to catch-up vaccines and will make sure everybody's up to date with DTP and MenACWY, even if they're not going to a country for which you need ACWY. GP 17

Travel vaccines were often given privately due to recommendations these should be done outside of the core GP contract, and this was primarily the case for adults but not children.

We do … Hepatitis A and then typhoid as part of the core contract. Anything else we direct patients to a private travel clinic. GP 24

However, there was variability in provision, with one GP stating: 'We don't charge for anything, including malaria pills' (GP 17). This would impact the 'migrant population who are going backwards and forwards to their home countries [and] constitute quite a large percentage of patients that we see for travel clinics' (GP 17).

Occupational vaccines were mentioned as sometimes being provided 'outside the schedule' for healthcare staff, such as nurses.

[Hepatitis B] vaccination is not within the schedule, so it has to be treated like a private prescription… some of them are nurses …[and they ] usually come from the South Indian population. Carers and nurses. GP 4

We shouldn't be seeing people wanting occupational health-related vaccination, but we do often get people asking for that. PN 1

### Strengthening vaccine delivery in UK primary care

Primary care staff raised a range of potential solutions and action points to increasing vaccine uptake, especially in adult migrants, including addressing personal, societal and physical barriers to vaccination systems through UK primary care alongside financial incentives to primary care to deliver adult catch-up vaccination. Key barriers and respective solutions identified by participants have been summarised in table 4.

## DISCUSSION
### Key findings

WHO's new Immunization Agenda 2030[9] calls for greater focus to be placed on delivering vaccination across the life course, targeting underimmunised groups for catch-up vaccination at any age, with primary care services therefore having a key role to play in the UK context. In our study, however, participants highlighted direct and indirect barriers to delivering catch-up vaccines in adult migrants who may have missed vaccines as children, missed boosters and not be aligned with the UK's vaccine schedule. Barriers were noted at a personal and service-delivery level, with themes including: lack of training and knowledge of guidance around catch-up vaccination among staff; unclear or incomplete vaccine records; and lack of incentivisation (including financial reimbursement), prioritisation and dedicated time and care pathways. Adult migrants were therefore reported as being excluded from many vaccination initiatives, most of which focus exclusively on children. In addition, PCPs reported that migrant patients express a range of views around vaccines to them, from positivity to uncertainty, to

| Table 4 | Barriers and solutions identified | | | |
|---|---|---|---|---|
| Barrier | Potential solution | Key messages | Quotes | Professional |
| Awareness of vaccination programmes for adults | Community engagement, capacity development, investment and partnership-building to raise awareness | Engage with community leaders, faith groups to help GPs and public health systems to improve uptake for vaccines in migrants; provide opportunities for information sharing, outreach, engagement, communication | 'I …hope that the CCG have thought about this and have gone to local communities, through the mosque or through other social avenues to trying get [vaccine] uptake.' | GP 24 |
| Fear of authorities | Community engagement to tackle mistrust; increasing trustworthiness of health and other institutions | Education and raising awareness within communities to overcome fear and enable health-seeking of preventative healthcare; (re)building trust through community engagement and investment | 'We have suggested … that they engage with the churches, that they obviously engage with information and advice, but it's a hard nut to crack if somebody's life is built around not trusting the specific institution.' | GP 24 |
| Misinformation about vaccines | Use trusted professionals or other trusted messengers—and ensure they are properly resourced, recognised and compensated (17) | One GP thought that consulting with someone who was felt to be an 'expert' in vaccinations would have better outcomes | '…If [the vaccine advice is from] from a GP…[or] from a consultant… then that tends to have a bit more weight to it… I think it depends on the level of education and understanding…' | GP 25 |
| | System approach—building capacity to recognise and respond to misinformation; developing resources to increase health literacy; | Public health messaging and a national approach | 'I think it's got to be a national approach… We got the Public Health Department…' | GP 22 |
| | Patient education; develop tailored messages | Patient education and sharing as much information as possible regarding vaccines, from all health professionals involved | 'People just need as much information as possible [about the vaccine], and I think information in particular on side effects etc.' | HCA 1 |
| Lack of training for staff around migrant health | Staff education and training (both clinical and non-clinical staff) | Improving staff understanding of potential issues and communication skills | 'It's just a bit of understanding… some patients may come across as difficult… [but with ] extra training with staff… [understanding can improve].' | HCA 2 |
| Financial pressures | Financial payments and incentives | Including adult migrant vaccination targets as a financial incentive to ensure migrant adult catch-up programmes are carried out | '…Unless they actually make [adult catch-up vaccination] something that they want GP surgeries to do, like proactively educate them and give them some renumeration to do it. work is money and we haven't got enough practice nurses as it is…So it can't just be expected to be an add on.' | GP 18 |
| Lack of time | Longer appointments | Longer appointments, especially if interpreter is needed | 'We make the appointments longer.' | PN 7 |

Continued

**Table 4** Continued

| Barrier | Potential solution | Key messages | Quotes | Professional |
|---|---|---|---|---|
| Language barrier | Interpreters; linguistically and culturally tailored information | Use interpreters for vaccine programmes, including written communication | 'We sent out a lot of text messages [about vaccination]. That would be good if we could do those in different languages…' | PN 15 |
| Different vaccine schedules and lack of history | Migrant specific health check | A health check for adult migrants, to gather information about vaccine history | '[A] template which is specific for patients from different countries, which means that you're not trawling through evidence.' | GP 20 |
| Pressures on health system | Ensure primary care deliver these vaccination programmes | Make migrant adult catch-up vaccination mandatory for primary care to provide | 'If they were part of QOF, they're made mandatory… that would definitely make [practices] do it.' | Pharmacist |

GP, general practitioner; HCA, healthcare assistants ; PN, practice nurse; QOF, Quality Outcomes Framework.

refusal. Some migrants including Somali, Eastern Europeans and Bangladeshi groups were often reported as being hesitant to get vaccinated, with specific concerns reported for specific vaccines, including MMR but with more positive responses to travel vaccinations. Greater consideration needs to be placed on potential delivery points for catch-up vaccination in adult migrants—for example, local places of worship and other trusted or familiar sites—alongside offering financial incentives or inclusion of adult migrant vaccination targets in QOF. Improving uptake of catch-up vaccination in this group will require new care pathways and training of frontline staff, alongside working directly with local community groups to communicate the benefits of vaccination at all ages. In addition, greater collaboration across systems and community groups and culturally competent campaigns are warranted. At a time when COVID-19 vaccination programmes are being rolled-out across the world, this study adds important understanding regarding the specific vaccination needs and concerns of migrants, and the challenges faced by the staff delivering vaccination programmes to migrant populations and older cohorts.

### Strengths and limitations
A key strength of the study is the number and variety of primary care staff included from across England in diverse settings. Interviewees were a self-selecting group, which may have affected the profile of those responding—a common consideration in qualitative research. However, a range of practices were involved, including those that do not see many migrants, and this diversity and the scale of the study is likely to have added to the breadth of experience and solutions reflected in our findings, as well as enhancing the validity. We noted that often participants made broad generalisations about specific nationality groups, which needs to be considered with commitment to equality, diversity and inclusion when assessing findings. The structure and experience of primary care across

Europe and between the devolved nations of the UK may differ so the recruitment only within England may limit the generalisability of the findings, however other European and international studies[7 17 18] have come to the similar conclusions in terms of healthcare provider, system and patient-related barriers to catch-up vaccination in relation to adult migrants, so we feel that this would be unlikely.

### Next steps for strengthening catch-up vaccination in older cohorts
We found a range of direct and indirect barriers to delivering catch-up vaccines in adult migrants who may have missed vaccines as children, missed boosters and not be aligned with the UK's vaccine schedule, from both a personal and service-delivery level. Our findings concur with those of similar study in Norway[17] which found no consistent or structured approach to vaccinating adult migrants in Norway, including no guidelines from governing bodies on how to organise vaccination to adult migrants. Reasons why adult vaccination is not prioritised included tuberculosis screening and treatment taking precedence, and a common assumption among healthcare providers that vaccinations are dealt with in childhood.[17] A questionnaire survey of experts across Europe,[7] and policy analysis,[19] found that policies and practice differ across European countries with respect to adult vaccination and the inclusion of migrants in vaccine systems on arrival. Only 13 of 32 countries in the European Union/European Economic Area (EU/EEA) had policies in place to offer MMR vaccines to adult migrants, with 10 countries reporting that they would charge fees.[7] Variations in vaccine policies targeting adult migrants were reported in another European survey.[20] In addition, it is well known that some migrants face a range of barriers to health systems more broadly. This suggests that more inclusive policies are required placing an emphasis on

new approaches to ensure older migrants are included, and that such policies are well implemented in practice.

Implementation will be key, and our study raised numerous points that merit greater consideration. Service delivery barriers have previously been described in other areas of migrant health, including screening for infection, with GPs citing concerns about lack of awareness around the health needs of migrants and insufficient time and resources.[21 22] It has previously been noted that negative biases from healthcare staff towards migrant patients or preconceptions about vaccine hesitancy in specific ethnic groups may have an impact on patient trust,[23 24] which is known to be a major factor in vaccine uptake.[25] Education and training of frontline providers will be a critical component given the critical role that the PCP–patient relationship has for building trust in vaccination. This must involve raising awareness of the diverse experiences of migrants and how to approach potential vaccination concerns with sensitivity, as well ensuring an understanding around the potential low vaccine coverage in their countries of origin as children, different dosing schedules and particularly low coverage for newer vaccines. For HPV, for example, global coverage for the final dose was only 13% in 2021[26]—suggesting many migrants aged under 25 years would be eligible for HPV vaccination as part of the UK's more advanced programme. However, likely a key factor will be financial incentivisation to encourage practices to target potentially underimmunised adults for catch-up vaccines, which was a recurrent theme among those interviewed. Catch-up vaccination could be considered at various entry points, for example, the New Patient Health Check or the NHS Health Check. Since April 2020, MMR now comes with an item of service payment, including for catch-up vaccination in patients who missed out on scheduled vaccines, which should encourage practices to offer appropriate vaccinations to patients regardless of age.

Tackling hesitancy and educating migrant and broader ethnic minority communities about the benefits of vaccination across the life course will also be a critical component,[22 27] with COVID-19 presenting numerous innovations in service delivery in this area that merit further consideration to routine vaccination going forward including outreach, policy shifts to facilitate registration of migrants with primary care providers and anonymous vaccination in trusted locations.[22 28] We found that certain nationality groups (Somali, Eastern Europeans and Bangladeshi) may be more hesitant to receive vaccines than others, or reluctant to receive certain vaccines, aligning with a recent systematic review that found nationality/country of origin to be a key determinant of vaccine uptake for routine vaccines and COVID-19 vaccines in European data sets.[8] In this study, acceptance barriers were mostly reported in Eastern European and Muslim migrants for HPV, measles and influenza vaccines, with 23 significant determinants of undervaccination in migrants found

(p<0.05), including African origin, recent migration and being a refugee/asylum seeker.[8]

A systematic review of interventions to improve vaccination uptake in newly-arrived migrants to the EU/EEA[29] highlighted the potential solutions of social mobilisation and outreach programmes, planned vaccinations and educational campaigns. Our data points to a recommendation for policymakers to include adult migrants especially in catch-up vaccination programmes on arrival, and to ensure policy around the delivery of catch-up vaccination across the life course is implemented in practice.

**Author affiliations**
¹Institute for Infection and Immunity, St George's University of London, London, UK
²LSHTM, London, UK
³Infection and Immunity Research Institute, and Population Health Research Institute, St George's University of London, London, UK
⁴UKHSA, London, UK
⁵Doctors of the World UK, London, UK
⁶Primary Care, Imperial College London, London, UK

**Acknowledgements** We thank members of our National Institute for Health Research Patient and Public Involvement Project Advisory Board (Babatunde Tikare, Larysa Agbaso, Monika Hartmann, Saliha Majeed) and additional members of our Project Board for valuable contributions.

**Contributors** JC and SH conceived the idea and developed the initial proposal. JC wrote the ethics applications, led the recruitment and data collection and contributed to data analysis and manuscript revision. AM led the data analysis and contributed to the manuscript draft, revision and concepts. FK contributed to data collection, data analysis, manuscript draft and revision. AD and AFC contributed to the data collection and manuscript revision. AM, JC, FK and SH wrote a first draft of the paper, with input from AFC, AD, LPG, FW, YC and AM. SH acts as guarantor for this work.

**Funding** This study was funded by the National Institute for Health Research (NIHR) and the Academy of Medical Sciences. FK is supported by a Health Education England/NIHR Academic Clinical Fellowship. JC is funded by an NIHR in-practice clinical fellowship (NIHR300290). The views expressed are those of the author(s) and not necessarily those of the National Health Service, the NIHR or the Department of Health and Social Care. AD and Sally E Hayward are supported by Medical Research Council PhD studentships (reference number: MR/ N013638/1). SH is funded by an NIHR Advanced Fellowship (reference number: NIHR300072), the Academy of Medical Sciences (reference number: SBF005\1111), the Novo Nordisk Foundation/La Caixa Foundation (Mobility–Global Medicine and Health Research grant) and the WHO. AFC is funded by the Academy of Medical Sciences (SBF005\1111) and the NIHR (NIHR300072).

**Competing interests** None declared.

**Patient and public involvement** Patients and/or the public were involved in the design, or conduct, or reporting, or dissemination plans of this research. Refer to the Methods section for further details.

**Patient consent for publication** Not applicable.

**Ethics approval** Ethics was granted by St George's, University of London Research Ethics Committee (2020.00630) and the Health Research Authority (REC 20/ HRA/1674). Participant information sheets were circulated, and signed informed consent was acquired prior to telephone interview. Participants consented to audio-recorded interviews. Participants gave informed consent to participate in the study before taking part.

**Provenance and peer review** Not commissioned; externally peer reviewed.

**Data availability statement** Data are available upon reasonable request.

**ORCID iDs**
Jessica Carter http://orcid.org/0000-0001-9590-3146
Alison F Crawshaw http://orcid.org/0000-0003-0450-7258
Lucy Pollyanna Goldsmith http://orcid.org/0000-0002-6934-1925
Sally Hargreaves http://orcid.org/0000-0003-2974-4348

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
