## [Reviewer comments · BMJ Open]

ARTICLE DETAILS

TITLE (PROVISIONAL)	"We don't routinely check vaccination background in adults": A national qualitative study of barriers and facilitators to vaccine delivery and uptake in adult migrants through UK primary care
AUTHORS	Carter, Jessica; Mehrotra, Anushka; Knights, Felicity; Deal, Anna; Crawshaw, Alison; Farah, Yasmin; Goldsmith, Lucy; Wurie, Fatima; Ciftci, Yusuf; Majeed, Azeem; Hargreaves, Sally

VERSION 1 – REVIEW

REVIEWER	Dutta, Tapati Fort Lewis College
REVIEW RETURNED	20-Apr-2022

GENERAL COMMENTS	Thank you for choosing me as a reviewer for this manuscript entitled "We don't routinely check vaccination background in adults": A national qualitative study of barriers and facilitators to vaccine delivery and uptake in adult migrants through UK primary care.' The manuscript is befitting the vision and scope of the BMJ Open journal and addresses a very relevant topic 'primary care professionals perspectives on vaccination catchups among adult migrants'. The article very effectively brings out some of the implementation level barriers and policy level recommendations. Overall, the manuscript is coherent, has the key stipulated sections Abstract, Background, Methods, and Discussions. I'd like to suggest major revisions and additional citations, towards strengthening this piece even more, and advancing being considered for publication. Some of the spellings are British English, e.g. 'immunised', versus 'immunized', or 'generalisability' versus 'generalizability'. But I would leave that to the journal's guidelines. Suggested edits are also mentioned as comments in the attached manuscript review. Title: I'd like to suggest deleting the comment "We don't routinely check vaccination background in adults". While that makes the title sounds catchy, not too sure if that complements anything to the later part of the title 'A national qualitative study of barriers and facilitators to vaccine delivery and uptake in adult migrants through UK primary care'. Rather, and to be more specific, I'd suggest rephrasing 'UK healthcare professionals' perspectives of barriers and facilitators for catch-up vaccinations among adult migrants' Abstract: In the Conclusions section "WHO's new Immunization Agenda (IA2030) called for greater focus to be placed on delivering vaccination across the life-course, targeting under-immunised groups for catch-up vaccination at any age, authors needs to
--

	clarify the phrase ‘increase perceived health’, ranking process, scale etc.” – I would suggest moving that to the Introduction section, in describing the policy context Keywords: Authors can consider replacing Europe with UK; and ‘Catchup adult vaccination’ rather than vaccination. The later one (Catchup adult vaccination) because its connotation and strategies are very different than regular vaccination. Also, the words ‘immunization’ and ‘vaccination’ cannot be used interchangeably. Strengths and limitations This part should highlight the unique strengths of the study- a nationwide qualitative study capturing healthcare professionals’ perspectives on catch-up vaccinations among adult migrants. Not too sure if this is the first of its kinds study on this topic? If so, please highlight that. No. (iii), the word ‘breadth’ is mentioned twice, please use another synonym. Methods: Design: Please describe the research team, including background, knowledgeability of the healthcare providers, time spent in building an amicable relationship with the respondent, such that good, honest and true data were available etc. Authors can also refer to the COREQ (CONSolidated criteria for Reporting Qualitative research) Checklist ISSM_COREQ_Checklist.pdf (elsevier.com) and report the same in the Methods section. Setting For the benefit of international audience, please add some details of the health care system and governance in the UK, as in how many rural, urban, or suburban centers, and what was the percentage of the settings from among which research participants participated? Participants All Participants who showed interest: Please detail out in a flowchart, how many potential participants were initially reached out, how many reached out through snowballing, how many declined, and the final list of participants. 20 Pounds: Were the respondents part of the Govt healthcare system? In some countries there is a norm that Govt folks cannot be incentivized to participate in research. Was curious to know about the system in the UK. Is any special permission needed to incentivize? Data Collection Transcribe: Possibly this will be more of a Data cleaning and analysis part, rather than ‘Data Collection’. Also, authors need to describe if researchers did the transcribing, or a software was used, or it was outsourced. Many times, in qualitative research, even these processes are deterministic to researcher’s reflexivity, and positionality. Results
--	---

	Table 1: Delete the word 'included'. The word Participants in itself means that they were included Table 4: Were the Solutions (in column 2) identified by the interviewees or the research team? Table 3 and 4 almost gives the same essence (limitations/ barriers), arranged differently. Can both the tables be combined 'barriers by vaccines,?' A similar table highlighting Enabling factors to vaccination needs to be added. That would justify having the same (Enabling factors) in the title. Finally Potential solution/ recommendation can be added as a separate narrative. QOF: Full form? Page 18: "I can remember.....". Can another quote be used here, because the same quote has been used in another place Even though this is a qualitative study, there is scope to add primary and secondary outcomes. Primary outcome can be the vaccination uptake/ vaccination catch up initiatives, while secondary outcomes can be the relational, and ethical aspects advanced or not advanced based on the interaction between the target populations and healthcare professionals. Discussion "We noted that often" noteworthy, and this part can be positioned using the Diversity, Equity, and Inclusion lens Adding these references to the relevant sections will be helpful and thus, recommended "Variations in vaccine policies...."  • Dutta, T., Meyerson, B., & Agle, J. (2018). African cervical cancer prevention and control plans: A scoping review. Journal of cancer policy, 16, 73-81. Perceived enablers and barriers for vaccination uptake:  • Dutta, T., Agle, J., Meyerson, B. E., Barnes, P. A., Sherwood-Laughlin, C., & Nicholson-Crotty, J. (2021). Perceived enablers and barriers of community engagement for vaccination in India: Using socioecological analysis. Plos one, 16(6), e0253318. • Dutta, T. (2019). Decision-makers' Conceptualization and Fostering of Community Engagement for Improved Adoption and Uptake of Existing and Emerging Vaccines in India (Doctoral dissertation, Indiana University). Life-course approach:  • Dutta, T., Agle, J., Lin, H. C., & Xiao, Y. (2021, May). Gender-responsive language in the National Policy Guidelines for Immunization in Kenya and changes in prevalence of tetanus vaccination among women, 2008–09 to 2014: A mixed methods study. In Women's Studies International Forum (Vol. 86, p. 102476). Pergamon.
--	--

REVIEWER	Mytton, Julie
----------	---------------

	University West of England, Centre for Child and Adolescent Health
REVIEW RETURNED	27-May-2022

GENERAL COMMENTS	Thank you for the opportunity to review this manuscript. The authors have identified that poor uptake of immunisation in migrant adults is an important issue, and that there is currently a paucity of evidence of why this is happening and what potential strategies could be implemented to address this issue. The study has the potential to contribute to knowledge, though it could be greatly strengthened by further consideration, particularly the presentation of the results. Title: Clear. Abstract: Good, structured as per guidance. The results section may need revision in response to the comments made below on this section. Introduction: Helpfully sets the scene, describes the drivers for improved vaccine uptake in adult migrants, and highlights the evidence gap. Methods: The inclusion of a migrant advisory board significantly strengthened the community engagement and involvement in this study, informing the development of materials that were subsequently piloted prior to use. Ethical approval was clear. It was unclear how the 50 practices were identified. The implication is that some may have been involved via their CRN but others appeared to be approached independently. In the section on data collection, it would be helpful to know how the interviews were distributed among the four researchers involved in data collection. (for example, was this geographical? or based on the profession of the participant? Random? etc). No mention is made of field notes being recorded until it is introduced with regard to the lost digital recordings of three participants. Results: This was the weakest part of the manuscript. Table 1 was helpful. Rather than reporting the mean age of participants, I would have been interested in the age range of participants and to know how many years the participants had been in their posts prior to recruitment to this study. A series of subheadings were used to break up the results, but it was not stated if these were the themes and if so, if all themes were reported (for example, training and lack of guidance are barely mentioned in the results yet are the first theme highlighted in the discussion). The results section could be strengthened by inclusion of an initial description of the themes and subthemes that were identified. As the focus of the study was on immunisation uptake in migrant adults, the inclusion of quotes relating to the uptake of immunisation in migrant children in Table 2 felt inappropriate. Decision making with regard to immunisation is likely to differ for a parent making a decision for their child compared with making a decision for themselves. The authors repeatedly referred to 'direct' and 'indirect' barriers but these terms were not defined. Language was described as an indirect barrier, but isn't lack of a common language a direct barrier? At the top of page 14, the authors state that 'mistrust of the NHS system generally' was an indirect barrier without further
--

	explanation or examples. Most of the text in this section related to language difficulties rather than trust issues. On page 15, in the section on barriers related to specific vaccines, it states that 'a summary of key themes, by vaccine, reported in Table 3' though Table 3 is titled 'key issues' and has a column titled 'key message'. This is a further example of how it is difficult for the reader to understand what themes were identified from the participants' manuscripts. Rather than reporting issues relating to specific vaccines, I would have thought that cross-vaccine issues would have been the important output to report since the aim of the study was not to improve the uptake of specific vaccines but to understand the barriers and facilitators to vaccine uptake in general. Table 4 was an interesting set of both barriers and solutions that were identified by individual participants. Presentation of the barriers is at times repetitive of what has already been presented in the 'barriers' section of the results. Rather than presenting a range of individual opinions of potential solutions to specific barriers it would be helpful to understand the cross-cutting issues / solutions that appear to be emerging, such as trust, communication, vaccine guidance, training, data systems, delivery systems etc etc. I wonder if results section would be easier for the reader, if it were structured as a series of paragraphs, each justifying the identification of a different barrier and then reporting the participants suggestions to address that barrier? However, without knowing what themes and subthemes were identified through the thematic analysis process it is difficult to know if this suggestion is appropriate. Discussion: Care should be taken to ensure consistency across the manuscript. In the first paragraph of the discussion, the authors state that "Somali, Eastern-Europeans and Bangladeshi groups were often reported as being hesitant to get vaccinated", whilst in the results section you have stated one PN reporting that Bangladeshi families are 'more engaged'. Overall, the discussion was well structured with good references to the existing literature. If the authors make changes to the results section in response to the feedback above they will need to determine the degree to which this necessitates revision of the discussion section.
--	--

VERSION 1 – AUTHOR RESPONSE

Reviewer Reports:

Reviewer: 1

Dr. Tapati Dutta, Fort Lewis College

Comments to the Author:

Thank you for choosing me as a reviewer for this manuscript entitled "'We don't routinely check vaccination background in adults": A national qualitative study of barriers and facilitators to vaccine delivery and uptake in adult migrants through UK primary care.' The manuscript is befitting the vision and scope of the BMJ Open journal and addresses a very relevant topic 'primary care professionals perspectives on vaccination catchups among adult migrants'. The article very effectively brings out some of the implementation level barriers and policy level recommendations. Overall, the manuscript is coherent, has the key stipulated sections Abstract, Background, Methods, and Discussions. I'd like

to suggest major revisions and additional citations, towards strengthening this piece even more, and advancing being considered for publication. Some of the spellings are British English, e.g. 'immunised', versus 'immunized', or 'generalisability' versus 'generalizability'. But I would leave that to the journal's guidelines.

Many thanks for these positive comments. Re: spelling, we have published in BMJ Open before, and it is English spelling. The WHO Immunization Agenda and World Health Organization are nouns and always spelt with a Z, so we have kept these.

Suggested edits are also mentioned as comments in the attached manuscript review (bmjopen-2022-062894_reviewed_4_22.pdf).

Many thanks we have taken these on board.

Title:

I'd like to suggest deleting the comment "We don't routinely check vaccination background in adults". While that makes the title sounds catchy, not too sure if that complements anything to the later part of the title 'A national qualitative study of barriers and facilitators to vaccine delivery and uptake in adult migrants through UK primary care'. Rather, and to be more specific, I'd suggest rephrasing 'UK healthcare professionals' perspectives of barriers and facilitators for catch-up vaccinations among adult migrants'

We are really keen to keep the title as it currently stands, which we feel captures the essence of the paper. The comment from the editor above, suggests that editorially the journal editors are happy with our decision on this point.

Abstract:

In the Conclusions section "WHO's new Immunization Agenda (IA2030) called for greater focus to be placed on delivering vaccination across the life-course, targeting under-immunised groups for catch-up vaccination at any age, authors needs to clarify the phrase 'increase perceived health', ranking process, scale etc." – I would suggest moving that to the Introduction section, in describing the policy context

We are confused about this point and can't see the words 'increased perceived health' in the conclusion section of the abstract. We have re-read the abstract and feel happy with it.

Keywords:

Authors can consider replacing Europe with UK; and 'Catchup adult vaccination' rather than vaccination. The later one (Catchup adult vaccination) because its connotation and strategies are very different than regular vaccination. Also, the words 'immunization' and 'vaccination' cannot be used interchangeably.

This has been done, many thanks for these comments.

Strengths and limitations

This part should highlight the unique strengths of the study- a nationwide qualitative study capturing healthcare professionals' perspectives on catch-up vaccinations among adult migrants. Not too sure if this is the first of its kinds study on this topic? If so, please highlight that.

Not sure to what extent this is the first, we are always reluctant to say that.

No. (iii), the word 'breadth' is mentioned twice, please use another synonym.

Thank-you for spotting this, we have reworked this sentence

Methods:

Design:

Please describe the research team, including background, knowledgeability of the healthcare providers, time spent in building an amicable relationship with the respondent, such that good, honest and true data were available etc. Authors can also refer to the COREQ (COnsolidated criteria for Reporting Qualitative research) Checklist ISSM_COREQ_Checklist.pdf (elsevier.com) and report the same in the Methods section.

We have adhered to COREQ and have added the following paragraph into the design/Methods section to describe the team: The team comprised two GPs and four academics and was supported by a wider project board of a diverse group of migrant ambassadors. The range of professional and personal experience supported integration of multiple perspectives throughout the design, collection and analysis stages. The inclusion of two GPs in the research team brought knowledge of UK primary care to the study but required careful reflection during interviews and data analysis and was balanced by the inclusion of non-GP research team members at the interview stage.

Setting

For the benefit of international audience, please add some details of the health care system and governance in the UK, as in how many rural, urban, or suburban centers, and what was the percentage of the settings from among which research participants participated?

Thank you for this comment we have added the below sentence to the study for clarification of this point

Participants were recruited from 50 GP practices with 50 participants were from practices (78%) from urban settings and 14 participants (22%) from suburban or rural settings across England

Participants

All Participants who showed interest: Please detail out in a flowchart, how many potential participants were initially reached out, how many reached out through snowballing, how many declined, and the final list of participants.

We did not record this information. We have published many qualitative studies in the past and not included a flow chart like this, but happy to generate something if the editor insists.

20 Pounds: Were the respondents part of the Govt healthcare system? In some countries there is a norm that Govt folks cannot be incentivized to participate in research. Was curious to know about the system in the UK. Is any special permission needed to incentivize?

Thank you for this question. All participants were NHS primary care employees, £20 gift voucher was offered to participants which has been common practice in research studies we have been part of and are aware of, this was approved by both local and NHS ethics boards with the references for this found in the paper.

Data Collection

Transcribe: Possibly this will be more of a Data cleaning and analysis part, rather than 'Data Collection'. Also, authors need to describe if researchers did the transcribing, or a software was used,

or it was outsourced. Many times, in qualitative research, even these processes are deterministic to researcher's reflexivity, and positionality.

Thank you for this comment the following sentence has been added to the paper for clarification. All but three of the interviews were digitally recorded and transcribed verbatim by professional transcription service.

Results

Table 1: Delete the word 'included'. The word Participants in itself means that they were included

Thank you for pointing this out this has been deleted

Table 4: Were the Solutions (in column 2) identified by the interviewees or the research team?

Barriers and solutions were identified by participants this has been clarified with addition of below sentence.

Key barriers and respective solutions identified by participants have been summarised in Table 4.

Table 3 and 4 almost gives the same essence (limitations/ barriers), arranged differently. Can both the tables be combined 'barriers by vaccines,'?

A similar table highlighting Enabling factors to vaccination needs to be added. That would justify having the same (Enabling factors) in the title.

Finally Potential solution/ recommendation can be added as a separate narrative.

Thank you for this comment, we had many discussions around presentation of this data as a research team and are very keen to keep the tables as they are following feedback from primary care colleagues of useful nature of these tables divided by specific vaccine and then with barriers and solutions presented together.

QOF: Full form?

This has been given in full form in abstract and earlier in paper and therefore acronym used from then on.

Page 18: "I can remember.....". Can another quote be used here, because the same quote has been used in another place

Thank you for identifying this we have replaced with the below illustrative quote.

"[on local community infectious disease led clinic] And they have a large Somalian support network there, so they have interpreters, and bits and pieces..... They will go in, and there will be a Somalian phlebotomist and doctor, and so they engage with it that way, much easier." HCA 6

Even though this is a qualitative study, there is scope to add primary and secondary outcomes. Primary outcome can be the vaccination uptake/ vaccination catch up initiatives, while secondary outcomes can be the relational, and ethical aspects advanced or not advanced based on the interaction between the target populations and healthcare professionals.

Thank you for this insight. We rarely do primary and secondary outcomes for qualitative studies but objectives. In our abstract, we have stated the following as two overarching objectives of the study,

repeated at the end of Intro to explain the key aims of this work: To explore the views of primary care professionals around barriers and facilitators to catch-up vaccination in adult migrants (defined as foreign born; over 18 years) with incomplete or uncertain vaccination status and for routine vaccines to inform development of future interventions to improve vaccine uptake in this group and improve coverage.

Discussion

"We noted that often" noteworthy, and this part can be positioned using the Diversity, Equity, and Inclusion lens

We agree that should be highlighted further and have added the below sentence

We noted that often participants made broad generalisations about specific nationality groups, which needs to be considered with commitment to equality, diversity and inclusion when assessing findings.

Adding these references to the relevant sections will be helpful and thus, recommended

Thank you for these interesting paper suggestions the following has been added to the introduction.

"Variations in vaccine policies...."

- Dutta, T., Meyerson, B., & Agle, J. (2018). African cervical cancer prevention and control plans: A scoping review. *Journal of cancer policy*, 16, 73-81.

Reviewer: 2

Dr. Julie Mytton, University West of England

Comments to the Author:

Thank you for the opportunity to review this manuscript. The authors have identified that poor uptake of immunisation in migrant adults is an important issue, and that there is currently a paucity of evidence of why this is happening and what potential strategies could be implemented to address this issue. The study has the potential to contribute to knowledge, though it could be greatly strengthened by further consideration, particularly the presentation of the results.

Thank you for this positive feedback.

Title: Clear.

Abstract: Good, structured as per guidance. The results section may need revision in response to the comments made below on this section.

Thank you for these comments

Introduction: Helpfully sets the scene, describes the drivers for improved vaccine uptake in adult migrants, and highlights the evidence gap.

Thank you for these positive comments

Methods: The inclusion of a migrant advisory board significantly strengthened the community engagement and involvement in this study, informing the development of materials that were subsequently piloted prior to use. Ethical approval was clear. It was unclear how the 50 practices were identified. The implication is that some may have been involved via their CRN but others appeared to be approached independently. In the section on data collection, it would be helpful to

know how the interviews were distributed among the four researchers involved in data collection. (for example, was this geographical? or based on the profession of the participant? Random? etc). No mention is made of field notes being recorded until it is introduced with regard to the lost digital recordings of three participants.

Thank you for these comments – recruitment is explained on page 10 of the manuscript under participants – please see below:

Recruitment occurred via local Clinical Research Networks, 'word of mouth' invitations from colleagues and a number of primary care newsletters, social media groups and practice manager mailing lists.

The below has been added for clarification of additional points

Telephone interviews, between 30-60 minutes, were carried out by JC (GP) FK, (GP registrar) and AD and AFC (academic researchers) who made field notes throughout. Interviews were distributed randomly to research team members.

Results: This was the weakest part of the manuscript. Table 1 was helpful. Rather than reporting the mean age of participants, I would have been interested in the age range of participants and to know how many years the participants had been in their posts prior to recruitment to this study.

Thank you for this comment we have added the below.

Participants were aged between 25 and 74 with a mean age of 45 years old (SD 11.8) and had been working in primary care between 1 and 35 years (mean 12.27 years SD 9.45).

A series of subheadings were used to break up the results, but it was not stated if these were the themes and if so, if all themes were reported (for example, training and lack of guidance are barely mentioned in the results yet are the first theme highlighted in the discussion). The results section could be strengthened by inclusion of an initial description of the themes and subthemes that were identified.

Many thanks for this comment we have inserted the following paragraph into the results for clarity:

Participants had varied exposure of vaccine delivery in migrant patients, but the data were convergent across this breadth of migrant healthcare experience, geographical area, and participant profession. The main themes that emerged from data analysis were; the existence of multiple barriers to the delivery of catch-up vaccination to migrant patients, including vaccine acceptance and PCP training; the fragmented nature of adult migrant catch-up vaccination models despite existence of guidelines; the role of travel vaccination and occupational health have in adult migrant catch-up vaccination and next steps for strengthening delivery of catch up vaccination with existence of positive attitudes to strengthening primary care's role through numerous PCP enacted or suggested solutions to barriers given.

As the focus of the study was on immunisation uptake in migrant adults, the inclusion of quotes relating to the uptake of immunisation in migrant children in Table 2 felt inappropriate. Decision making with regard to immunisation is likely to differ for a parent making a decision for their child compared with making a decision for themselves.

Although the focus was adult immunisation, often adult immunisation is discussed in the context of childhood immunisation, and we found overlap that we felt was useful. For example, migrants views on whether or not they would vaccinate their children with MMR and their views on MMR we felt was

relevant to whether or not they would themselves have a catch-up MMR as an adult, with MMR the key vaccine for adult catch-up. We discussed this within the team and decided to keep data related to children within the paper do feel table 2 is new and interesting data that summarises the views of these communities about their views on vaccination as adults. We think this is useful data to include but we take on board the concerns of the reviewer and have removed 1 of the quotes in table 2 for balance.

The authors repeatedly referred to 'direct' and 'indirect' barriers but these terms were not defined. Language was described as an indirect barrier, but isn't lack of a common language a direct barrier? At the top of page 14, the authors state that 'mistrust of the NHS system generally' was an indirect barrier without further explanation or examples. Most of the text in this section related to language difficulties rather than trust issues.

Thank-you for these useful comments. We will remove focus on direct and indirect and describe them all as "barriers" as we agree with the reviewer this is a little confusing and have included all identified barriers as subheadings with descriptive quotes under overall theme of multiple barriers for clarity which we feel adds to the flow of the results section.

On page 15, in the section on barriers related to specific vaccines, it states that 'a summary of key themes, by vaccine, reported in Table 3' though Table 3 is titled 'key issues' and has a column titled 'key message'. This is a further example of how it is difficult for the reader to understand what themes were identified from the participants' manuscripts. Rather than reporting issues relating to specific vaccines, I would have thought that cross-vaccine issues would have been the important output to report since the aim of the study was not to improve the uptake of specific vaccines but to understand the barriers and facilitators to vaccine uptake in general.

Thank you for this helpful comment, we have clarified the above and renamed the table and description as barriers per specific vaccine. The section above outlines the cross-cutting vaccine issues but following discussion as a team we feel that the specific vaccine barriers in table 3 represent useful data in particular at a clinical level as many vaccine programmes are vaccine specific.

Table 4 was an interesting set of both barriers and solutions that were identified by individual participants. Presentation of the barriers is at times repetitive of what has already been presented in the 'barriers' section of the results. Rather than presenting a range of individual opinions of potential solutions to specific barriers it would be helpful to understand the cross-cutting issues / solutions that appear to be emerging, such as trust, communication, vaccine guidance, training, data systems, delivery systems etc etc. I wonder if results section would be easier for the reader, if it were structured as a series of paragraphs, each justifying the identification of a different barrier and then reporting the participants suggestions to address that barrier? However, without knowing what themes and subthemes were identified through the thematic analysis process it is difficult to know if this suggestion is appropriate.

Thank-you for these positive comments on table 4. We are keen to keep this table in as it presents barriers next to potential solutions which is useful for those using the data to assist in designing new vaccine delivery interventions, and we were keen to keep the solutions in a separate section as we think it works a little better for flow and really is a key focus of our paper.

Thank-you for the comments on this section we now feel the results is clearer with changes made above where we have clearly now highlighted the 4 key thematic areas in the introduction and each section is a theme. We think these changes have improved flow for this section.

Discussion: Care should be taken to ensure consistency across the manuscript. In the first paragraph of the discussion, the authors state that “Somali, Eastern-Europeans and Bangladeshi groups were often reported as being hesitant to get vaccinated”, whilst in the results section you have stated one PN reporting that Bangladeshi families are ‘more engaged’.

Thank you for this comment, we understand the confusion as the PN reporting engagement was in relation to travel vaccination, we have rephrased this for clarification.

Some migrants including Somali, Eastern-Europeans and Bangladeshi groups were often reported as being hesitant to get vaccinated, with specific concerns reported for specific vaccines, including MMR but with more positive responses to travel vaccinations.

Overall, the discussion was well structured with good references to the existing literature. If the authors, make changes to the results section in response to the feedback above they will need to determine the degree to which this necessitates revision of the discussion section.

Thank you for the positive response to the discussion, we have revisited the discussion post-results changes and now feel the results and discussion sections complement each other.

Reviewer: 1

Competing interests of Reviewer: No competing interests

Reviewer: 2

Competing interests of Reviewer: No competing interests to declare

VERSION 2 – REVIEW

REVIEWER	Dutta, Tapati Fort Lewis College
REVIEW RETURNED	16-Sep-2022

GENERAL COMMENTS	Thanks for making substantial changes in the revised version of the manuscript and addressing all the review comments. Some more references are suggested towards augmentation: FOR COMMUNITY ENGAGEMENT TO ENHANCE VACCINATION Dutta, T., Agle, J., Meyerson, B. E., Barnes, P. A., Sherwood-Laughlin, C., & Nicholson-Crotty, J. (2021). Perceived enablers and barriers of community engagement for vaccination in India: Using socioecological analysis. Plos one, 16(6), e0253318. Dutta, T., Meyerson, B. E., Agle, J., Barnes, P. A., Sherwood-Laughlin, C., & Nicholson-Crotty, J. (2020). A qualitative analysis of vaccine decision makers’ conceptualization and fostering of ‘community engagement’ in India. International journal for equity in health, 19(1), 1-14. FOR CULTURE RESPONSIVE VACCINATION MESSAGING Dutta, T., Agle, J., Lin, H. C., & Xiao, Y. (2021, May). Gender-responsive language in the National Policy Guidelines for Immunization in Kenya and changes in prevalence of tetanus vaccination among women, 2008–09 to 2014: A mixed methods
--

	study. In Women's Studies International Forum (Vol. 86, p. 102476). Pergamon.
--	---